# LEARNING FROM NOISY DATA WITH ROBUST REPRESENTATION LEARNING

## ABSTRACT

Learning from noisy data has attracted much attention, where most methods focus on label noise. In this work, we propose a new framework which simultaneously addresses three types of noise commonly seen in real-world data: label noise, out-of-distribution input, and input corruption. In contrast to most existing methods, we combat noise by learning robust representation. Specifically, we embed images into a low-dimensional subspace by training an autoencoder on the deep features. We regularize the geometric structure of the subspace with robust contrastive learning, which includes an unsupervised consistency loss and a supervised mixup prototypical loss. Furthermore, we leverage the structure of the learned subspace for noise cleaning, by aggregating information from neighboring samples. Experiments on multiple benchmarks demonstrate state-of-the-art performance of our method and robustness of the learned representation. Our code will be released[1].

## 1 INTRODUCTION

Data in real life is *noisy*. However, deep models with remarkable performance are mostly trained on clean datasets with high-quality human annotations. Manual data cleaning and labeling is an expensive process that is difficult to scale. On the other hand, there exists almost infinite amount of noisy data online. It is crucial that deep neural networks (DNNs) could harvest noisy training data. However, it has been shown that DNNs are susceptible to overfitting to noise (Zhang et al., 2017).

As shown in Figure 1, a real-world noisy image dataset often consists of multiple types of noise. *Label noise* refers to samples that are wrongly labeled as another class (*e.g.* flower labeled as orange). *Out-of-distribution input* refers to samples that do not belong to any known classes. *Input corruption* refers to image-level distortion (*e.g.* low brightness) that causes data shift between training and test.

Most of the methods in literature focus on addressing the more detrimental label noise. Two dominant approaches include: (1) find clean samples as those with smaller loss and assign larger weights to them (Han et al., 2018; Yu et al., 2019; Shen & Sanghavi, 2019; Arazo et al., 2019); (2) relabel noisy samples using model's predictions (Reed et al., 2015; Ma et al., 2018; Tanaka et al., 2018; Yi & Wu, 2019). The recently proposed DivideMix (Li et al., 2020a) integrates both approaches in a co-training framework, but it also increases computation cost. Previous methods that focus on addressing label noise do not consider out-of-distribution input or input corruption, which limits their performance in real-world scenarios. Furthermore, using a model's own prediction to relabel samples could cause confirmation bias, where the prediction error accumulates and harms performance.

We propose a new direction for effective learning from noisy data. Our method embeds images into noise-robust low-dimensional representations, and regularizes the geometric structure of the representations with contrastive learning. Specifically, our **algorithmic contributions** include:

- We propose noise-robust contrastive learning, which introduces two contrastive losses. The first is an unsupervised consistency contrastive loss. It enforces inputs with perturbations to have similar normalized embeddings, which helps learn robust and discriminative representation.

- Our second contrastive loss is a weakly-supervised mixup prototypical loss. We compute class prototypes as normalized mean embeddings, and enforces each sample's embedding to be closer to

---

[1]Code is in the supplementary material

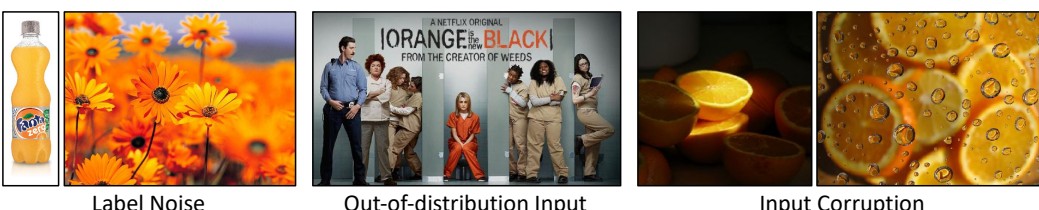

| Label Noise | Out-of-distribution Input | Input Corruption |

Figure 1: Google search images from WebVision (Li et al., 2017) dataset with keyword "orange".

its class prototype. Inspired by Mixup (Zhang et al., 2018), we construct virtual training samples as linear interpolation of inputs, and encourage the same linear relationship *w.r.t* the class prototypes.

- We train a linear autoencoder to reconstruct the high-dimensional features using low-dimensional embeddings. The autoendoer enables the high-dimensional features to maximally preserve the robustness of the low-dimensional embeddings, thus regularizing the classifier.

- We propose a new noise cleaning method which exploits the structure of the learned representations. For each sample, we aggregate information from its top-$k$ neighbors to create a pseudo-label. A subset of training samples with confident pseudo-labels are selected to compute the weakly-supervised losses. This process can effectively clean both label noise and out-of-distribution (OOD) noise.

Our **experimental contributions** include:

- We experimentally show that our method is robust to label noise, OOD input, and input corruption. Experiments are performed on multiple datasets with controlled noise and real-world noise, where our method achieves **state-of-the-art** performance.

- We demonstrate that the proposed noise cleaning method can effectively clean a majority of label noise. It also learns a curriculum that gradually leverages more samples to compute the weakly-supervised losses as the pseudo-labels become more accurate.

- We validate the robustness of the learned low-dimensional representation by showing (1) $k$-nearest neighbor classification outperforms the softmax classifier. (2) OOD samples can be separated from in-distribution samples. The efficacy of the proposed autoencoder is also verified.

## 2    RELATED WORK

**Label noise learning.** Learning from noisy labels have been extensively studied in the literature. While some methods require access to a small set of clean samples (Xiao et al., 2015; Vahdat, 2017; Veit et al., 2017; Lee et al., 2018; Hendrycks et al., 2018), most methods focus on the more challenging scenario where no clean labels are available. These methods can be categorized into two major types. The first type performs label correction using predictions from the network (Reed et al., 2015; Ma et al., 2018; Tanaka et al., 2018; Yi & Wu, 2019). The second type tries to separate clean samples from corrupted samples, and trains the model on clean samples (Han et al., 2018; Arazo et al., 2019; Jiang et al., 2018; 2020; Wang et al., 2018; Chen et al., 2019; Lyu & Tsang, 2020). The recently proposed DivideMix (Li et al., 2020a) effectively combines label correction and sample selection with the Mixup (Zhang et al., 2018) data augmentation under a co-training framework. However, it cost $2\times$ the computational resource of our method.

Different from existing methods, our method combats noise by learning noise-robust low-dimensional representations. We propose a more effective noise cleaning method by leveraging the structure of the learned representations. Furthermore, our model is robust not only to label noise, but also to out-of-distribution and corrupted input. A previous work has studied open-set noisy labels (Wang et al., 2018), but their method does not enjoy the same level of robustness as ours.

**Contrastive learning.** Contrastive learning is at the core of recent self-supervised representation learning methods (Chen et al., 2020; He et al., 2019; Oord et al., 2018; Wu et al., 2018). In self-supervised contrastive learning, two randomly augmented images are generated for each input image. Then a contrastive loss is applied to pull embeddings from the same source image closer, while pushing embeddings from different source images apart. Recently, prototypical contrastive learning (PCL) (Li et al., 2020b) has been proposed, which uses cluster centroids as prototypes, and trains the network by pulling an image embedding closer to its assigned prototypes.

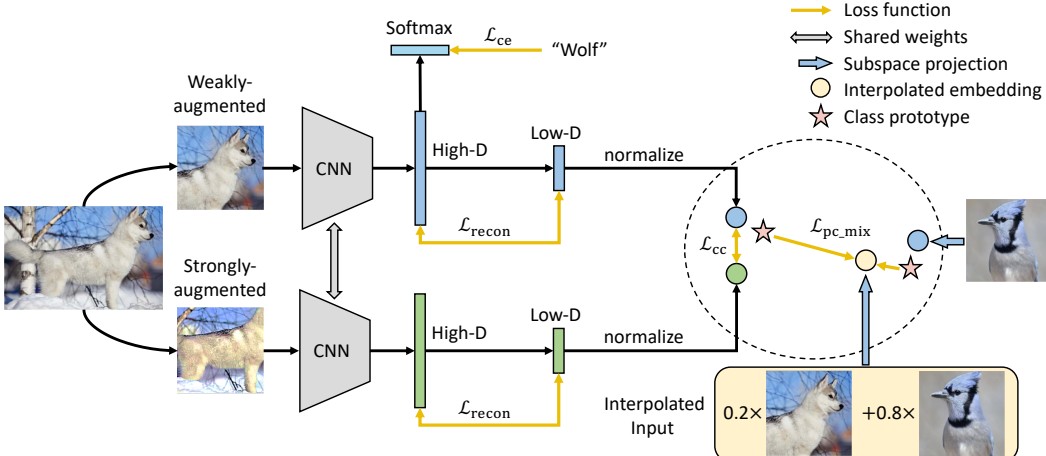

Figure 2: Our proposed framework for noise-robust contrastive learning. We project images into a low-dimensional subspace, and regularize the geometric structure of the subspace with (1)$\mathcal{L}_{cc}$ a consistency contrastive loss which enforces images with perturbations to have similar embeddings; (2)$\mathcal{L}_{pc\_mix}$: a prototypical contrastive loss augmented with mixup, which encourages the embedding for a linearly-interpolated input to have the same linear relationship *w.r.t* the class prototypes. The low-dimensional embeddings are also trained to reconstruct the high-dimensional features, which preserves the learned information and regularizes the classifier.

Different from previous methods, our method performs contrastive learning in the principal subspace of the high-dimensional feature space, by training a linear autoencoder. Furthermore, our supervised contrastive loss improves PCL (Li et al., 2020b) with Mixup (Zhang et al., 2018). Different from the original Mixup where learning happens at the classification layer, our learning takes places in the low-dimensional subspace.

## 3 METHOD

Given a noisy training dataset $\mathcal{D} = \{(\boldsymbol{x}_i, y_i)\}_{i=1}^n$, where $\boldsymbol{x}_i$ is an image and $y_i \in \{1, ..., C\}$ is its class label. We aim to train a network that is robust to the noise in training data (*i.e.* label noise, OOD input, input corruption) and achieves high accuracy on a clean test set. The proposed network consists of three components: (1) a deep encoder (a convolutional neural network) that encodes an image $\boldsymbol{x}_i$ to a high-dimensional feature $\boldsymbol{v}_i$; (2) a classifier (a fully-connected layer followed by softmax) that receives $\boldsymbol{v}_i$ as input and outputs class predictions; (3) a linear autoencoder that projects $\boldsymbol{v}_i$ into a low-dimensional embedding $\boldsymbol{z}_i \in \mathbb{R}^d$. We show an illustration of our method in Figure 2, and a pseudo-code in appendix B. Next, we delineate its details.

### 3.1 CONTRASTIVE LEARNING IN ROBUST LOW-DIMENSIONAL SUBSPACE

Let $\boldsymbol{z}_i = \mathbf{W}_e \boldsymbol{v}_i$ be the linear projection from high-dimensional features to low-dimensional embeddings, and $\hat{\boldsymbol{z}}_i = \boldsymbol{z}_i / \|\boldsymbol{z}_i\|_2$ be the normalized embeddings. We aim to learn robust embeddings with two contrastive losses: unsupervised consistency loss and weakly-supervised mixup prototypical loss.

**Unsupervised consistency contrastive loss**. Following the NT-Xent (Chen et al., 2020) loss for self-supervised representation learning, our consistency contrastive loss enforces images with semantic-preserving perturbations to have similar embeddings. Specifically, given a minibatch of $b$ images, we apply weak-augmentation and strong-augmentation to each image, and obtain $2b$ inputs $\{\boldsymbol{x}_i\}_{i=1}^{2b}$. Weak augmentation is a standard flip-and-shift augmentation strategy, while strong augmentation consists of color and brightness changes with details given in Section 4.1.

We project the inputs into the low-dimensional space to obtain their normalized embeddings $\{\hat{\boldsymbol{z}}_i\}_{i=1}^{2b}$. Let $i \in \{1, ..., b\}$ be the index of a weakly-augmented input, and $j(i)$ be the index of the strong-

augmented input from the same source image, the consistency contrastive loss is defined as:

$$\mathcal{L}_{\text{cc}} = \sum_{i=1}^{b} -\log \frac{\exp(\hat{\boldsymbol{z}}_i \cdot \hat{\boldsymbol{z}}_{j(i)}/\tau)}{\sum_{k=1}^{2b} \mathbb{1}_{i \neq k} \exp(\hat{\boldsymbol{z}}_i \cdot \hat{\boldsymbol{z}}_k/\tau)}, \tag{1}$$

where $\tau$ is a scalar temperature parameter. The consistency contrastive loss maximizes the inner product between the pair of positive embeddings $\hat{\boldsymbol{z}}_i$ and $\hat{\boldsymbol{z}}_{j(i)}$, while minimizing the inner product between $2(b-1)$ pairs of negative embeddings. By mapping different views (augmentations) of the same image to neighboring embeddings, the consistency contrastive loss encourages the network to learn discriminative representation that is robust to low-level image corruption.

**Weakly-supervised mixup prototypical contrastive loss**. Our second contrastive loss injects structural knowledge of classes into the embedding space. Let $\mathcal{I}_c$ denote indices for the subset of images in $\mathcal{D}$ labeled with class $c$, we calculate the class prototype as the normalized mean embedding:

$$\boldsymbol{z}^c = \frac{1}{|\mathcal{I}_c|} \sum_{i \in \mathcal{I}_c} \hat{\boldsymbol{z}}_i, \quad \hat{\boldsymbol{z}}^c = \frac{\boldsymbol{z}^c}{\|\boldsymbol{z}^c\|_2}, \tag{2}$$

where $\hat{\boldsymbol{z}}_i$ is the embedding of a center-cropped image, and the class prototypes are calculated at the beginning of each epoch.

The prototypical contrastive loss enforces an image embedding $\hat{\boldsymbol{z}}_i$ to be more similar to its corresponding class prototype $\hat{\boldsymbol{z}}^{y_i}$, in contrast to other class prototypes:

$$\mathcal{L}_{\text{pc}}(\hat{\boldsymbol{z}}_i, y_i) = -\log \frac{\exp(\hat{\boldsymbol{z}}_i \cdot \hat{\boldsymbol{z}}^{y_i}/\tau)}{\sum_{c=1}^{C} \exp(\hat{\boldsymbol{z}}_i \cdot \hat{\boldsymbol{z}}^c/\tau)}. \tag{3}$$

Since the label $y_i$ is noisy, we would like to regularize the encoder from memorizing training labels. Mixup (Zhang et al., 2018) has been shown to be an effective method against label noise (Arazo et al., 2019; Li et al., 2020a). Inspired by it, we create virtual training samples by linearly interpolating a sample (indexed by $i$) with another sample (indexed by $m(i)$) randomly chosen from the same minibatch:

$$\boldsymbol{x}_i^m = \lambda \boldsymbol{x}_i + (1 - \lambda)\boldsymbol{x}_{m(i)}, \tag{4}$$

where $\lambda \sim \text{Beta}(\alpha, \alpha)$.

Let $\hat{\boldsymbol{z}}_i^m$ be the normalized embedding for $\boldsymbol{x}_i^m$, the mixup version of the prototypical contrastive loss is defined as a weighted combination of the two $\mathcal{L}_{\text{pc}}$ *w.r.t* class $y_i$ and $y_{m(i)}$. It enforces the embedding for the interpolated input to have the same linear relationship *w.r.t.* the class prototypes.

$$\mathcal{L}_{\text{pc\_mix}} = \sum_{i=1}^{2b} \lambda \mathcal{L}_{\text{pc}}(\hat{\boldsymbol{z}}_i^m, y_i) + (1 - \lambda)\mathcal{L}_{\text{pc}}(\hat{\boldsymbol{z}}_i^m, y_{m(i)}). \tag{5}$$

**Reconstruction loss**. We also train a linear decoder $\mathbf{W}_{\text{d}}$ to reconstruct the high-dimensional feature $\boldsymbol{v}_i$ based on $\boldsymbol{z}_i$. The reconstruction loss is defined as:

$$\mathcal{L}_{\text{recon}} = \sum_{i=1}^{2b} \|\boldsymbol{v}_i - \mathbf{W}_{\text{d}}\boldsymbol{z}_i\|_2^2. \tag{6}$$

There are several benefits for training the autoencoder. First, with an optimal linear autoencoder, $\mathbf{W}_{\text{e}}$ will project $\boldsymbol{v}_i$ into its low-dimensional principal subspace and can be understood as applying PCA (Baldi & Hornik, 1989). Thus the low-dimensional representation $\boldsymbol{z}_i$ is intrinsically robust to input noise. Second, minimizing the reconstruction error is maximizing a lower bound of the mutual information between $\boldsymbol{v}_i$ and $\boldsymbol{z}_i$ (Vincent et al., 2010). Therefore, knowledge learned from the proposed contrastive losses can be maximally preserved in the high-dimensional representation, which helps regularize the classifier.

**Classification loss**. Given the softmax output from the classifier, $\boldsymbol{p}(\boldsymbol{y}; \boldsymbol{x}_i)$, we define the classification loss as the cross-entropy loss. Note that it is only applied to the weakly-augmented inputs.

$$\mathcal{L}_{\text{ce}} = -\sum_{i=1}^{b} \log p(y_i; \boldsymbol{x}_i). \tag{7}$$

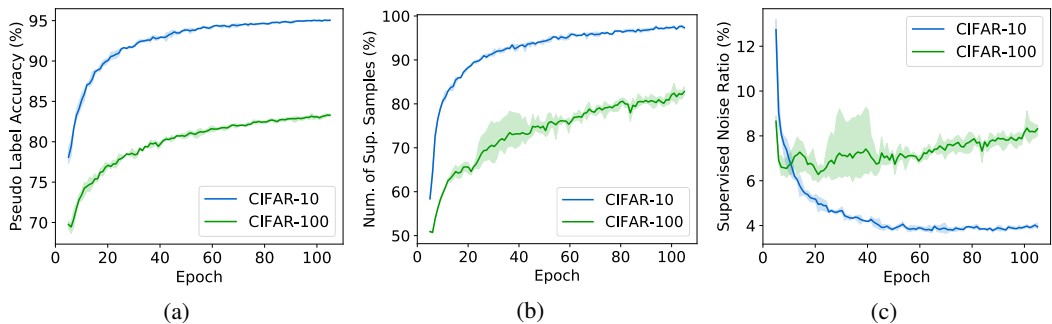

Figure 3: Curriculum learned by the proposed label correction method for training on CIFAR datasets with 50% sym. noise. (a) Accuracy of pseudo-labels *w.r.t* to clean training labels. (b) Number of samples in the weakly-supervised subset $\mathcal{D}_{\text{sup}}^t$. (c) Label noise ratio in the weakly-supervised subset.

The overall training objective is to minimize a weighted sum of all losses:

$$\mathcal{L} = \mathcal{L}_{\text{ce}} + \omega_{\text{cc}}\mathcal{L}_{\text{cc}} + \omega_{\text{pc}}\mathcal{L}_{\text{pc\_mix}} + \omega_{\text{recon}}\mathcal{L}_{\text{recon}} \quad (8)$$

For *all* experiments, we fix $\omega_{\text{cc}} = 1$, $\omega_{\text{recon}} = 1$, and change $\omega_{\text{pc}}$ only across datasets.

## 3.2 NOISE CLEANING WITH SMOOTH NEIGHBORS

After warming-up the model by training with the noisy labels $\{y_i\}_{i=1}^n$ for $t_0$ epochs, we aim to clean the noise by generating a soft pseudo-label $q_i$ for each training sample. Different from previous methods that perform label correction purely using the model's softmax prediction, our method exploits the structure of the low-dimensional subspace by aggregating information from top-$k$ neighboring samples, which helps alleviate the confirmation bias problem.

At the $t$-th epoch, for each sample $x_i$, let $p_i^t$ be the classifier's softmax prediction, let $q_i^{t-1}$ be its soft label from the previous epoch, we calculate the soft label for the current epoch as:

$$q_i^t = \frac{1}{2}p_i^t + \frac{1}{2}\sum_{j=1}^k w_{ij}^t q_j^{t-1}, \quad (9)$$

where $w_{ij}^t$ represents the normalized affinity between a sample and its neighbor and is defined as $w_{ij}^t = \frac{\exp(\hat{z}_i^t \cdot \hat{z}_j^t / \tau)}{\sum_{j=1}^k \exp(\hat{z}_i^t \cdot \hat{z}_j^t / \tau)}$. We set $k = 200$ in all experiments.

The soft label defined by eqn.(9) is the minimizer of the following quadratic loss function:

$$J(q_i^t) = \sum_{j=1}^k w_{ij}^t \left\| q_i^t - q_j^{t-1} \right\|_2^2 + \left\| q_i^t - p_i^t \right\|_2^2. \quad (10)$$

The first term is a smoothness constraint which encourages the soft label to take a similar value as its neighbors' labels, whereas the second term attempts to maintain the model's class prediction.

We construct a weakly-supervised subset which contains (1) *clean* sample whose soft label score for the original class $y_i$ is higher than a threshold $\eta_0$, (2) *pseudo-labeled* sample whose maximum soft label score exceeds a threshold $\eta_1$. For pseudo-labeled samples, we convert their soft labels into hard labels by taking the class with the maximum score.

$$\mathcal{D}_{\text{sup}}^t = \{x_i, y_i \mid q_i^t(y_i) > \eta_0\} \cup \{x_i, \hat{y}_i^t = \arg\max_c q_i^t(c) \mid \forall \max_c q_i^t(c) > \eta_1, c \in \{1, .., C\}\} \quad (11)$$

Given the weakly-supervised subset, we modify the classification loss $\mathcal{L}_{\text{ce}}$, the mixup prototypical contrastive loss $\mathcal{L}_{\text{pc\_mix}}$, and the calculation of prototypes $\hat{z}^c$, such that they only use samples from $\mathcal{D}_{\text{sup}}^t$. The unsupervised losses (*i.e.* $\mathcal{L}_{\text{cc}}$ and $\mathcal{L}_{\text{recon}}$) still operate on all training samples.

**Learning curriculum.** Our iterative noise cleaning method learns an effective training curriculum, which gradually increases the size of $\mathcal{D}_{\text{sup}}^t$ as the pseudo-labels become more accurate. To demonstrate

| Dataset | CIFAR-10 | | | CIFAR-100 | |
|---|---|---|---|---|---|
| Noise type | Sym 20% | Sym 50% | Asym 40% | Sym 20% | Sym 50% |
| Cross-Entropy (Li et al., 2020a) | 82.7 | 57.9 | 72.3 | 61.8 | 37.3 |
| Forward (Patrini et al., 2017) | 83.1 | 59.4 | 83.1 | 61.4 | 37.3 |
| Co-teaching+ (Yu et al., 2019) | 88.2 | 84.1 | - | 64.1 | 45.3 |
| Mixup (Zhang et al., 2018) | 92.3 | 77.6 | - | 66.0 | 46.6 |
| P-correction (Yi & Wu, 2019) | 92.0 | 88.7 | 88.1 | 68.1 | 56.4 |
| MLNT (Li et al., 2019) | 92.0 | 88.8 | 88.6 | 67.7 | 58.0 |
| M-correction (Arazo et al., 2019) | 93.8 | 91.9 | 86.3 | 73.4 | 65.4 |
| DivideMix (Li et al., 2020a) | 95.0 | 93.7 | 91.4 | 74.8 | 72.1 |
| DivideMix (reproduced) | 95.1±0.1 | 93.6±0.2 | 91.3±0.8 | 75.1±0.2 | 72.1±0.3 |
| Ours (classifier) | 95.8±0.1 | 94.3±0.2 | 91.9±0.8 | 79.1±0.1 | 74.8±0.4 |
| Ours (knn) | **95.9**±0.1 | **94.5**±0.1 | **92.4**±0.9 | **79.4**±0.1 | **75.0**±0.4 |

Table 1: Comparison with state-of-the-art methods on CIFAR datasets with label noise. Numbers indicate average test accuracy (%) over last 10 epochs. We report results over 3 independent runs with randomly-generated label noise. Results for previous methods are copied from Arazo et al. (2019); Li et al. (2020a). We re-run DivideMix (without ensemble) using the publicly available code on the same noisy data as ours.

such curriculum, we analyse the noise cleaning statistics for training our model on CIFAR-10 and CIFAR-100 datasets with 50% label noise (experimental details explained in the next section). In Figure 3 (a), we show the accuracy of the soft pseudo-labels *w.r.t* to clean training labels (only used for analysis purpose). Our method can significantly reduce the ratio of label noise from 50% to 5% (for CIFAR-10) and 17% (for CIFAR-100). Figure 3 (b) shows the size of $\mathcal{D}_{\mathrm{sup}}^t$ as a percentage of the total number of training samples, and Figure 3 (c) shows the effective label noise ratio within the weakly-supervised subset $\mathcal{D}_{\mathrm{sup}}^t$. Our method maintains a low noise ratio in the weakly-supervised subset, while gradually increasing its size to utilize more samples for the weakly-supervised losses.

## 4 EXPERIMENT

In this section, we validate the proposed method on multiple benchmarks with controlled noise and real-world noise. Our method achieves state-of-the-art performance across all benchmarks. For fair comparison, we compare with DivideMix (Li et al., 2020a) without ensemble. In appendix A, we report the result of our method with co-training and ensemble, which further improves performance.

### 4.1 EXPERIMENTS ON CONTROLLED NOISY LABELS

**Dataset.** Following Tanaka et al. (2018); Li et al. (2020a), we corrupt the training data of CIFAR-10 and CIFAR-100 (Krizhevsky & Hinton, 2009) with two types of label noise: *symmetric* and *asymmetric*. Symmetric noise is injected by randomly selecting a percentage of samples and changing their labels to random labels. Asymmetric noise is class-dependant, where labels are only changed to similar classes (*e.g.* dog↔cat, deer→horse). We experiment with multiple noise ratios: sym 20%, sym 50%, and asym 40% (see results for sym 80% and 90% in appendix A). Note that asymmetric noise ratio cannot exceed 50% because certain classes would become theoretically indistinguishable.

**Implementation details.** Same as previous works (Arazo et al., 2019; Li et al., 2020a), we use PreAct ResNet-18 (He et al., 2016) as our encoder model. We set the dimensionality of the bottleneck layer as $d = 50$. Our model is trained using SGD with a momentum of 0.9, a weight decay of 0.0005, and a batch size of 128. The network is trained for 200 epochs. We set the initial learning rate as 0.02 and use a cosine decay schedule. We apply standard crop and horizontal flip as the weak augmentation. For strong augmentation, we use AugMix (Hendrycks et al., 2020), though other methods (*e.g.* SimAug (Chen et al., 2020)) work equally well. For all CIFAR experiments, we fix the hyper-parameters as $\omega_{\mathrm{cc}} = 1, \omega_{\mathrm{pc}} = 5, \omega_{\mathrm{recon}} = 1, \tau = 0.3, \alpha = 8, \eta_1 = 0.9$. For CIFAR-10, we activate noise cleaning at epoch $t_0 = 5$, and set $\eta_0 = 0.1$ (sym.) or 0.4 (asym.). For CIFAR-100, we activate noise cleaning at epoch $t_0 = 15$, and set $\eta_0 = 0.02$. We use faiss-gpu (Johnson et al., 2017) for efficient knn search in the low-dimensional subspace, which finishes within 1 second.

**Results.** Table 1 shows the comparison with existing methods. Our method outperforms previous methods across all label noise settings. On the more challenging CIFAR-100, we achieve 3-4% accuracy improvement compared to the second-best method DivideMix. Moreover, our method is more computational efficient than DivideMix, which needs co-training for noise filtering.

| CIFAR-10 50% sym. noise | CE | Iterative (Wang et al., 2018) | GCE (Zhang & Sabuncu, 2018) | DivideMix (Li et al., 2020a) | Ours (cls.) | Ours (knn) |
|---|---|---|---|---|---|---|
| + CIFAR-100 20k | 53.6 | 87.2 | 87.3 | 89.0 | 91.5 | **93.1**±0.3 |
| + SVHN 20k | 58.1 | 88.6 | 88.8 | 91.9 | 93.3 | **93.9**±0.2 |
| + Image Corruption | 53.8 | 87.7 | 87.9 | 89.8 | 91.4 | **91.6**±0.2 |

Table 2: Comparison with state-of-the-art methods on datasets with label noise and input noise. Numbers indicate average test accuracy (%) over last 10 epochs. We report results over 3 independent runs with randomly-generated noise. We re-run previous methods using publicly available code with the same noisy data and model architecture as ours.

In order to demonstrate the advantage of the proposed low-dimensional embeddings, we perform $k$-nearest neighbor (knn) classification ($k = 200$), by projecting test images into normalized embeddings. Compared to the trained classifier, knn achieves higher accuracy, which verifies the robustness of the learned low-dimensional representations.

## 4.2 EXPERIMENTS ON CONTROLLED NOISY LABELS WITH NOISY IMAGES

**Dataset.** We further corrupt a noisy CIFAR-10 dataset (sym. 50%) by injecting two types of input noise: out-of-distribution (OOD) images and input corruption. For OOD noise, we follow Wang et al. (2018) and add $20k$ images from either one of the two other datasets: CIFAR-100 and SVHN (Netzer et al., 2011), enlarging the training set to $70k$. A random CIFAR-10 label is assigned to each OOD image. For input corruption, we follow Hendrycks & Dietterich (2019) and corrupt each image in CIFAR-10 with a noise randomly chosen from the following four types: *Fog*, *Snow*, *Motion blur* and *Gaussian noise*. Examples of both types of input noise are shown in Figure 4. We follow the same implementation details as the CIFAR-10 experiments described in Section 4.1.

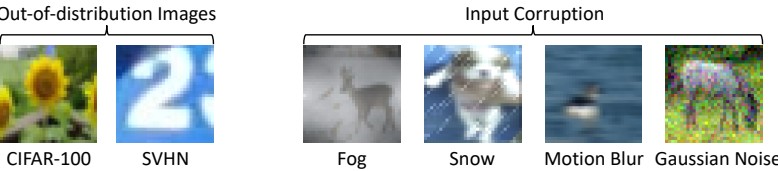

Figure 4: Examples of input noise injected to CIFAR-10.

**Results.** Table 2 shows the results, where our method consistently outperforms existing methods by a substantial margin. We observe that OOD images from a similar domain (CIFAR-100) are more harmful than OOD images from a more different domain (SVHN). This is because noisy images that are closer to the test data distribution are more likely to distort the decision boundary in a way that negatively affects test performance. Nevertheless, performing knn classification using the learned embeddings demonstrates high robustness to input noise.

In Figure 5, we show the t-SNE (Maaten & Hinton, 2008) visualization of the low-dimensional embeddings for all training samples. As training progresses, our model learns to separate OOD samples (represented as gray points) from in-distribution samples, and cluster samples of the same class together despite their noisy labels.

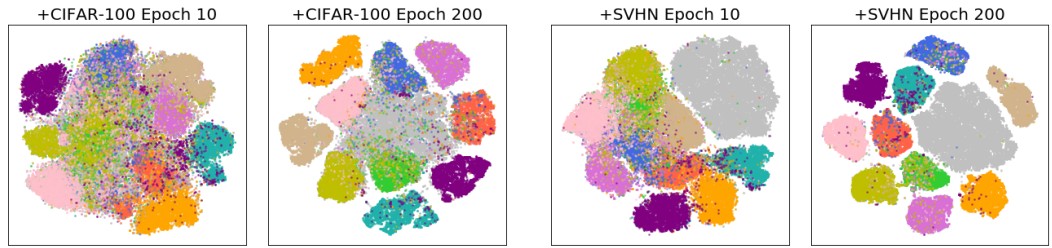

Figure 5: t-SNE visualization of low-dimensional embeddings for CIFAR-10 images (color represents the true class) + OOD images (gray points) from CIFAR-100 or SVHN. The model is trained on noisy CIFAR-10 ($50k$ images with 50% label noise) and $20k$ OOD images with random labels. Our method can effectively learn to (1) cluster CIFAR-10 images according to their true class, despite their noisy labels; (2) separate OOD samples from in-distribution samples, such that their harm is reduced.

| Test dataset | WebVision | | ILSVRC12 | |
|---|---|---|---|---|
| Accuracy (%) | top1 | top5 | top1 | top5 |
| Forward (Patrini et al., 2017) | 61.1 | 82.7 | 57.4 | 82.4 |
| Decoupling (Malach & Shalev-Shwartz, 2017) | 62.5 | 84.7 | 58.3 | 82.3 |
| D2L (Ma et al., 2018) | 62.7 | 84.0 | 57.8 | 81.4 |
| MentorNet (Jiang et al., 2018) | 63.0 | 81.4 | 57.8 | 79.9 |
| Co-teaching (Han et al., 2018) | 63.6 | 85.2 | 61.5 | 84.7 |
| INCV (Chen et al., 2019) | 65.2 | 85.3 | 61.0 | 85.0 |
| DivideMix (Li et al., 2020a) | 75.9 | 90.1 | 73.3 | 89.2 |
| Ours (w/o noise cleaning) | 75.5 | 90.2 | 72.0 | 90.0 |
| Ours (classifier) | 76.3 | **91.5** | 73.3 | **91.2** |
| Ours (knn) | **77.8** | 91.3 | **74.4** | 90.9 |

Table 3: Comparison with state-of-the-art methods trained on WebVision (mini).

| Method | CE | Forward | Joint-Opt | MLNT | MentorMix | SL | DivideMix | Ours (cls.) | Ours (knn) |
|---|---|---|---|---|---|---|---|---|---|
| Accuracy | 69.21 | 69.84 | 72.16 | 73.47 | 74.30 | 74.45 | 74.48 | 74.84 | **74.97** |

Table 4: Comparison with state-of-the-art methods on Clothing1M dataset.

## 4.3 EXPERIMENTS ON REAL-WORLD NOISY DATA

**Dataset and implementation details.** We verify our method on two real-word noisy datasets: WebVision (Li et al., 2017) and Clothing1M (Xiao et al., 2015). Webvision contains images crawled from the web using the same concepts from ImageNet ILSVRC12 (Deng et al., 2009). Following previous works (Chen et al., 2019; Li et al., 2020a), we perform experiments on the first 50 classes of the Google image subset. Clothing1M consists of images collected from online shopping websites where labels were generated from surrounding texts. Note that we do not use the additional clean set for training. For both experiments, we use the same model architecture as previous methods. More implementation details are given in the appendix.

**Results.** We report the results for WebVision in Table 3 and Clothing1M in Table 4, where we achieve state-of-the-art performance on both datasets. Our method achieves competitive performance on WebVision even *without* performing noise cleaning, which demonstrates the robustness of the learned representation. Appendix D shows examples of noisy images that are cleaned by our method.

## 4.4 ABLATION STUDY

**Effect of the proposed components.** In order to study the effect of the proposed components, we remove each of them and report accuracy of the classifier (knn) across four benchmarks. As shown in Table 5, the mixup prototypical contrastive loss ($\mathcal{L}_{pc\_mix}$) is most crucial to the model's performance. The consistency contrastive loss ($\mathcal{L}_{cc}$) has a stronger effect with corrupted input or larger number of classes. We also experiment with removing mixup and using the standard prototypical contrastive loss, and using standard data augmentation (crop and horizontal flip) instead of AugMix. The proposed method still achieves state-of-the-art result with standard data augmentation.

| | CIFAR-10 Sym 50% | + CIFAR-100 20k | + Image Corruption | CIFAR-100 Sym 50% |
|---|---|---|---|---|
| w/o $\mathcal{L}_{pc\_mix}$ | 85.9 (86.1) | 79.7 (81.5) | 81.6 (81.7) | 65.6 (65.9) |
| w/o $\mathcal{L}_{cc}$ | 93.7 (93.8) | 91.3 (91.5) | 89.4 (89.5) | 71.9 (71.8) |
| w/o $\mathcal{L}_{recon}$ | 93.3 (94.0) | 90.7 (92.9) | 90.2 (91.0) | 73.2 (73.9) |
| w/o mixup | 89.5 (89.9) | 85.4 (87.0) | 84.7 (84.9) | 69.3 (69.7) |
| w/ standard aug. | 94.1 (94.3) | 90.8 (92.9) | 90.5 (90.7) | 74.5 (75.0) |
| DivideMix | 93.6 | 89.0 | 89.8 | 72.1 |
| Ours | 94.3 (94.5) | 91.5 (93.1) | 91.4 (91.6) | 74.8 (75.0) |

Table 5: Effect of the proposed components. We show the accuracy of the classifier (knn) on four benchmarks with different noise. Note that DivideMix (Li et al., 2020a) also performs mixup.

**Effect of bottleneck dimension.** We vary the dimensionality of the bottleneck layer, $d$, and examine the performance change in Table 6. Our model is in general not very sensitive to the change of $d$.

| bottleneck dimension | $d = 25$ | $d = 50$ | $d = 100$ | $d = 200$ |
|---|---|---|---|---|
| CIFAR-10 Sym 50% | 93.4 | 94.3 | 94.2 | 93.7 |
| CIFAR-100 Sym 50% | 73.8 | 74.8 | 74.4 | 73.8 |

Table 6: Classifier's test accuracy (%) with different low-dimensions.

## 5  CONCLUSION

This paper proposes noise-robust contrastive learning, a new method to combat noise in training data by learning robust representation. We demonstrate our model's state-of-the-art performance with extensive experiments on multiple noisy datasets. For future work, we are interested in adapting our method to other domains such as NLP or speech. We would also like to explore the potential of our method for learning transferable representations that could be useful for down-stream tasks.

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
