# OpenReview forum: "Learning from Noisy Data with Robust Representation Learning"
_ICLR.cc/2021/Conference — Reject_

### Official Review · AnonReviewer2 · 2020-10-27
**This paper introduces a noise-robust contrastive learning loss. The authors first learn robust representation to regularize the learning procedure with label noise. Then, a noise cleaning process is implemented to select some confident samples for partially finetuning.**

**Rating:** 6
**Confidence:** 4

**Review:**

The authors have conducted a range of experiments to validate the performance of the proposed method. And according to the results, it is appealing that the proposed method achieves relative improvement compared to current SOTAs . However, there are some concerns as follows.

1) It is quite ad-hoc about the proposed method, which consists of multiple losses and two phases in the training progresses. The potential hyperparameter space is relative black-box to understand where is the bottleneck of learning with noisy labels in this method. It also requires careful hyperparameter-setting as shown in the implementation details.

2) Some contributions are over-claimed since many components have also used in previous works for learning with noisy labels, e.g., the idea of consistency loss and autoencoder in Bootstrapping [1].  Besides, the title of the paper is also misleading since it is about learning with noisy labels instead of contrastive learning.


[1] S. Reed et al. Training deep neural networks on noisy labels with bootstrapping.

---

> ### Author Response · Authors · 2020-11-16
> **Response to Reviewer #2**
>
> We appreciate the reviewer’s valuable feedback. Here are our responses to the concerns.
>
> 1. Concern on the complexity of the method.
> \
> We would like to highlight that our method can still achieve state-of-the-art results if some components and hyper-parameters are simplified. In our ablation study in Table 5, we show that our method achieves competitive or better performance than current SoTA when either the unsupervised contrastive loss or the reconstruction loss is removed. We also perform experiments using a fixed set of loss weights ($w_{cc}$=1, $w_{pc}$=2, $w_{recon}$=1), and still achieve SoTA performance. Please refer to our response to Reviewer 3 for detailed numbers. Furthermore, our proposed method is more computationally efficient than the current SoTA DivideMix, which requires co-training and ensemble of two models.
>
> 2. Concerns on the title and technical claims.
> \
> We thank the reviewer for the suggestion and have modified our title as “Learning from noisy data with robust representation learning”.  Different from existing methods, our method is the first to explore the direction of robust representation learning for learning from noisy data. The “unsupervised consistency contrastive loss” in our method is fundamentally different from the consistency loss in Bootstrapping. The Bootstrapping method is more closely related to our pseudo-labels, but we propose a better pseudo-labelling method by aggregating information from nearest neighbors.
> We would also like to emphasize that our method not only addresses label noise, but also out-of-distribution noise and input corruption, which commonly occur in real-world noisy datasets.
>
> Thanks again for your review. Please let us know if we have addressed your concerns or if you have other questions.

---

### Official Review · AnonReviewer3 · 2020-10-28
**Review for "Noise-Robust Contrastive Learning"**

**Rating:** 6
**Confidence:** 4

**Review:**

The authors of the paper propose to use the contrastive loss, the mixup prototypical loss, and a reconstruction loss to regularize the learned representation in order to achieve robustness under various kinds of noise like label noise, out-of-distribution input, and input corruption. A noise-cleaning process based on the learned representation is also introduced to further enhance the results. Extensive experiments were conducted to demonstrate the effectiveness of the method.

All in all, the paper is well written and easy to understand. The proposed method is well justified and seems to be technically sound. I also find the idea of using neighboring samples to perform noise-cleaning intuitive and interesting. A major complaint I have regarding this submission is the lack of novelty. While the empirical results are encouraging, the proposed method is merely a combination of various previously proposed methods. Moreover, the final objective used (Equation 8) also involves 3 hyper-parameters that can be hard to tune. How sensitive are the results when different weights are used? Why is the $w_{pc}$ changed across different datasets? Due to the limitation addressed above, I think the paper a very borderline submission.

Questions:
1. It was demonstrated in the appendix that the prototypical loss is the most critical of all the losses used in this paper. However, no ablation study was done with respect to the effect of the mixup on the prototypical loss. How would regular prototypical loss perform without mixup in the context of these experiments?
2. It was stated that the labels for the pseudo-labeled samples were converted into hard labels. Does this lead to better performance than if soft labels are used?
----------------------------------------------------------------------
After Rebuttal:

I would like to thank the authors for answering my questions and addressing my concerns on hyperparameters. I also appreciate the authors' efforts in the additional ablation study conducted. While I still do think that the novelty of the paper is a little bit lacking, I think the experiments are carefully conducted and the empirical results seem to be encouraging. As such, I have raised my score to 6.

---

> ### Author Response · Authors · 2020-11-16
> **Response to Reviewer #3**
>
> We appreciate the reviewer’s positive comments and valuable feedback. Here are our responses to the concerns. We would like to highlight that our method can still achieve state-of-the-art results if some components and hyper-parameters are kept simpler, as shown below. In terms of technical novelty, our method is the first to explore the direction of representation learning for learning from noisy data. Learning robust representation enables (1) better classifier performance, (2) better noise cleaning with nearest neighbors, and (3) better robustness to out-of-distribution samples and input corruption. To highlight this novelty, we have modified our title as “Learning from noisy data with robust representation learning”.
>
> 1. The final objective involves three additional losses. How sensitive are the results when different weights are used?
> \
> Our method is in general not sensitive to the weights of losses. In Table 5, we show that removing either the unsupervised contrastive loss or the reconstruction loss still yields performance competitive or better than the current SoTA.
>
> 2. Why is the $w_{pc}$ changed across different datasets?
> \
> To address this concern, we show that using the same $w_{pc}$ = 2 still achieves SoTA performance across all datasets. On WebVision, it has the same results as reported. On Clothing1M, it achieves an accuracy of 74.76 and 74.83 for classifier and knn, respectively. On CIFAR-10/100, the results are:
> | Method |CIFAR-10 Sym 20%|CIFAR-10 Sym 50%|CIFAR-10 Asym 40%|CIFAR-100 Sym 20%|CIFAR-100 Sym 50%|
> | ------------- |:-------------:|:-------------:| :-------------:|:-------------:|:-------------:|
> Classifier | 95.2 | 94.0 | 91.5 | 78.3 | 73.1 |
> Knn | 95.4 | 94.3 | 91.9 | 78.6 | 73.4 |
> DivideMix (Li et al., 2020a) | 95.1 | 93.6 | 91.3 | 75.1 | 72.1|
>
> 3. How would regular prototypical loss perform without mixup?
> \
> Following the valuable suggestion, we have performed additional ablation experiments where mixup is removed and the regular prototypical contrastive loss is used. The results are updated in Table 5 in the main text. Using the prototypical contrastive loss (4th row) leads to a substantial increase compared to the 1st row, but lower than our full method due to the removal of mixup. Note that the best existing method, DivideMix, also uses mixup as an important component.
>
> 4. Does hard labels lead to better performance than if soft labels are used?
> \
> Using hard labels does lead to better performance than soft labels. We conjecture that the reason is due to the effect of entropy minimization by hard labels, where the classifier’s prediction is encouraged to be more confident. To verify this hypothesis, we added a sharpening function to the soft pseudo-labels similar as in DivideMix. With the sharpening, we are able to achieve comparable or even better results than using hard labels. However, since it introduces an additional temperature hyperparameter, we do not use sharpening in our final method.
>
> Thanks again for your review. Please let us know if we have addressed your concerns or if you have other questions.

---

### Official Review · AnonReviewer1 · 2020-10-28
**An extention of contrastive learning for training noise-robust deep networks.**

**Rating:** 6
**Confidence:** 3

**Review:**

#######################################################################

Summary:

The paper proposes noise-robust contrastive learning to combat label noise, out-of-distribution input and input corruption simultaneously. In particular, this paper embeds images into low-dimensional representations by training an autoencoder, and regularizes the geometric structure of the representations by contrastive learning. Furthermore, this paper introduces a new noise cleaning method based on the structure of the representations. Training samples with confident pseudo-labels are selected for supervised learning to clean both label noise and out-of-distribution noise. The effectiveness of the proposed method has been evaluated on multiple simulated and real-world noisy datasets.

#######################################################################

Reasons for score:

Overall, I vote for a weak acceptance. The proposed noise-robust contrastive learning introduces two contrastive losses: unsupervised consistency loss and supervised mixup prototypical loss. My major concern is about the clarity of the paper and some additional issues (see cons below). Hopefully the authors can address my concern in the rebuttal period.

#######################################################################

Pros:

1. The paper takes one import issue in deep learning: learning from noisy data.

2. For me, the proposed supervised mixup prototypical contrastive loss is novel for learning with noisy data. Specifically, it injects structure knowledge of classes into embedding space by combining the mixup technique and prototypical contrastive loss. The design is reasonable and interesting.

3. This paper provides comprehensive experiments, including both qualitative analysis and quantitative results, to show the effectiveness of the proposed framework. In particular, the proposed method outperforms several state-of-the-art robust learning methods in learning with label noise, out-of-distribution input and input corruption.

#######################################################################

Cons:

1. For the motivation, it would be better to provide more details about it, which seems not very clear to me. Particularly, it is unclear why the contrastive loss is duly used in the paper. Will the functionality of the unsupervised contrastive loss be achieved in the supervised prototypical loss? Additionally, the prototypical contrastive loss in equation (1) is an InfoNCE with normalized mean embeddings as the prototypes, which seems different from the ProtoNCE in the original paper [1]. It is better to clarify the the differences of the formulation and training strategy, and the reason of design of supervised prototypical loss in this paper.

[1] Junnan Li, Pan Zhou, Caiming Xiong, Richard Socher, and Steven C.H. Hoi. Prototypical Contrastive Learning of Unsupervised Representations, In ICLR, 2020.

2. As the key contribution of this paper, mixup prototypical contrastive loss combines mixup technique and prototypical contrastive loss. In the appendix, the authors have provided ablation study to show the effect of proposed losses, and it shows that the mixup prototypical loss is most crucial to the model’s performance. Here, the authors utilizes mixup in two ways: first is to create virtual training samples, and the other is to define the mixup version of prototypical contrastive loss as an weighted combination of two prototypical contrastive loss with respect to true class and virtual class. However, the effect of mixup augmentation, prototypical contrastive loss and mixup prototypical contrastive loss are unclear.  Since mixup has been shown to be an effective method against label noise, it would be more convincing if the authors can study the individual effect of each components of the proposed loss in the rebuttal period.

3. The proposed method uses many data augmentations, e.g., standard crop and horizontal flip as weak augmentation and AugMix as strong augmentation in the unsupervised consistency contrastive loss, and mixup technique in the supervised prototypical contrastive loss. I am concerning about the fairness in the experimental comparison. It is unclear to me if the authors have applied the same data augmentation to all the compared methods.

#######################################################################

Questions during rebuttal period:

Please address and clarify the cons above.

---

> ### Author Response · Authors · 2020-11-16
> **Response to Reviewer #1**
>
> We appreciate the reviewer’s positive comments and valuable feedback. Here are our responses to the concerns:
>
> 1. Motivation of using contrastive learning.
> \
> Different from most existing works in learning from noisy data, our method addresses this problem by learning robust representation. Better representation enables (1) better classifier performance, (2) better noise cleaning with nearest neighbors, (3) better robustness to out-of-distribution samples and input corruption.  Contrastive learning has been shown as an effective approach for representation learning, thus the motivation.
> 2. Will the functionality of the unsupervised contrastive loss be achieved in the supervised prototypical loss?
> \
> Firstly, we have revised “supervised prototypical loss” into “weakly-supervised prototypical loss” since the supervision comes from weak pseudo-labels.
> \
> The unsupervised and weakly-supervised contrastive loss play different roles in shaping the geometry of the embedding space.  Specifically, the unsupervised consistency contrastive loss maps different augmentations of the same image to neighboring embeddings, and enforces different images to have different embeddings. It encourages the network to learn discriminative representation that is robust to low-level image corruption and ood samples. On the other hand, the weakly-supervised prototypical contrastive loss injects class-structural knowledge into the embedding space. It reduces the intra-class variance by clustering samples from the same class around the prototype, and increases the inter-class variance by pushing a sample away from the prototypes of negative classes.
> 3. Clarification of the formulation of the contrastive loss.
> \
> Eq. (1) describes the unsupervised contrastive loss that is the same as the InfoNCE loss in SimCLR. Eq. (3) describes the weakly-supervised prototypical contrastive loss that is a variation of the ProtoNCE loss in PCL (Li et al. 2020b). Different from the unsupervised ProtoNCE where prototypes are computed as cluster centroids, we compute a prototype for each class as the normalized averaged embedding for all samples whose pseudo-label predict this class.
> 4. Decouple the effect of mixup and prototypical contrastive loss.
> \
> Following the reviewer’s valuable suggestion, we have performed additional ablation experiments where mixup is removed and the regular prototypical contrastive loss is used. The results are updated in Table 5 in the main text. Using the prototypical contrastive loss (4th row) leads to a substantial increase compared to the 1st row, but lower than our full method due to the removal of mixup. Note that the best existing method, DivideMix, also uses mixup as an important component.
> 5. Results using standard augmentation.
> \
> Following the reviewer’s valuable suggestion, we have performed experiments with the standard crop-and-flip augmentation instead of AugMix. The results are updated in Table 5. Despite some slight decrease in accuracy, our method with standard augmentation still achieves state-of-the-art performance.
>
> Thanks again for your review. Please let us know if we have addressed your concerns or if you have other questions.

---

### Official Review · AnonReviewer4 · 2020-10-28
**An interesting combination of Mixup, SimCLR,  denoising autoencoding and noise-cleaning.**

**Rating:** 7
**Confidence:** 4

**Review:**

This paper proposes a new methodology for noisy-robust learning by augmenting the simlr methodology with a Mixup-style augmentation and noise-cleaning.

The paper proposes a new methodology for training models in the presence of label noise. The method is a combination of standard SimCLR: using standard image augmentations and enforcing a contrastive consistence loss between the emeddings of the weakly augmented and strongly augmented versions of the same image. In addition, to the standard augmentation, the method also uses a Mixup style augmentation for the strongly augmented images: that is the projection of a randomly selected image is combined (using a convex combination) with the original image and then another contrastive loss is enforced for the class prototyle.
The high dimensional embedding of the weakly augmented image is trained with the (noisy) label using a cross-entropy loss.
In addition, the high dimensional embedding is also reconstructed from the low-dimensional projection.

The final evaluation also contains a noise-cleaning step by generating pseudo-labels from the smooth-neighborhoods using the above embeddings.

The quality of the writing is high and the paper presents a plausible combination of several strong methods. The paper is nicely written, well motivated and can be followed easily.

The paper also presents significant improvement of artificially noisy versions of CIFAR-10 and CIFAR-100. Also it is also measured on versions on which the images are noised by using extra datasets (SVHN) to create more interesting augmentations of the datasets.

The weakness of the paper is that it is only measured on cifar-10/100 and these datasest are often not very representative of real noisy datasets with label noise of complicated structure.

Another weakness is that this paper presents a relatively complex approach composed of 4 different methods, each of them are well studied and with very limited ablation analyses. It is plausible that the combined approach should do well, still it has limited novelty as it is the straightforward combination of four known approaches also the paper does not give a disciplined overview and study of the contributions of the different components.

---

> ### Author Response · Authors · 2020-11-16
> **Response to Reviewer #4**
>
> We appreciate the reviewer’s valuable feedback. Here are our responses to the concerns:
>
> In Table 3 and Table 4, we have shown experimental results on two real-world noisy datasets: WebVision and Clothing1M. The proposed method achieves state-of-the-art performance on both datasets.
>
> In the original paper, we have performed ablation study in the appendix to demonstrate the contributions of different components. With the allowance of an additional page, we are able to move it to Table 5 in the main text. Following the reviewer’s suggestion, we also enrich the ablation study with more experiments. Out of the proposed components, the mixup prototypical contrastive loss is the most critical. Our method is able to achieve competitive performance when some other components are removed or simplified. Notably, without the reconstruction loss, our method’s performance decreases by ~1%, but is still the state-of-the-art.
>
> Here we emphasize the role of the three proposed losses in learning noise-robust representation, which is also explained in the paper:
> 1. The consistency contrastive loss maps different augmentations of the same image to neighboring embeddings, and enforces different images to have different embeddings. It encourages the network to learn discriminative representation that is robust to low-level image corruption and out-of-distribution samples.
> 2. The prototypical contrastive loss injects class-structural knowledge into the embedding space by clustering samples around their corresponding class prototypes. The mixup regularization enforces the embedding of an interpolated input to be similar to two class prototypes.
> 3. The reconstruction loss enables the high-dimensional features to maximally preserve the robustness of the low-dimensional embeddings.
>
> Thanks again for your review. Please let us know if we have addressed your concerns or if you have other questions.

---

### Decision · Program_Chairs · 2021-01-07
**Final Decision**

**Decision:**

Reject

**Comment:**

This paper proposes a framework to train a discriminative model robust against (i) label noise, (ii) out-of-distribution input, and (iii) input corruption. To tackle these problems, a complex model is proposed that combines several existing models including InfoNCE-style contrastive learning, prototypical contrastive loss, Mixup, and reconstruction loss. Noisy training labels are cleaned using a temporally consistent label smoothing mechanism, combined with a curriculum learning algorithm.

Originally, the reviewers raised concerns regarding the limited ablation experiments and the lack of studies on real-world noisy labels. The additional experiments in the revised version addressed some of these concerns. Thus, the reviewers increased their rating slightly.

However, the reviewers in the discussion phase agree that the proposed method has a limited novelty, is complex, and involves many moving parts that require a careful design and hyperparameter tuning, and they do not recommend accepting the submission. I agree with the reviewers and recommend rejection.